# The Multiple Roles and Therapeutic Potential of Molecular Chaperones in Prostate Cancer

**DOI:** 10.3390/cancers11081194

**Published:** 2019-08-16

**Authors:** Abdullah Hoter, Sandra Rizk, Hassan Y. Naim

**Affiliations:** 1Department of Biochemistry and Chemistry of Nutrition, Faculty of Veterinary Medicine, Cairo University, 12211 Giza, Egypt; 2Department of Physiological Chemistry, University of Veterinary Medicine Hannover, 30559 Hannover, Germany; 3School of Arts and Sciences, Lebanese American University, Beirut 1102 2801, Lebanon

**Keywords:** prostate cancer, therapeutic resistance, heat shock proteins (HSPs), molecular chaperones, HSPs inhibitors, biomarkers

## Abstract

Prostate cancer (PCa) is one of the most common cancer types in men worldwide. Heat shock proteins (HSPs) are molecular chaperones that are widely implicated in the pathogenesis, diagnosis, prognosis, and treatment of many cancers. The role of HSPs in PCa is complex and their expression has been linked to the progression and aggressiveness of the tumor. Prominent chaperones, including HSP90 and HSP70, are involved in the folding and trafficking of critical cancer-related proteins. Other members of HSPs, including HSP27 and HSP60, have been considered as promising biomarkers, similar to prostate-specific membrane antigen (PSMA), for PCa screening in order to evaluate and monitor the progression or recurrence of the disease. Moreover, expression level of chaperones like clusterin has been shown to correlate directly with the prostate tumor grade. Hence, targeting HSPs in PCa has been suggested as a promising strategy for cancer therapy. In the current review, we discuss the functions as well as the role of HSPs in PCa progression and further evaluate the approach of inhibiting HSPs as a cancer treatment strategy.

## 1. Introduction

Prostate cancer (PCa) is amongst the high-risk cancers for males, especially in old age. Although Western countries show the highest incidence rates of prostate tumors, recent reports imply growing numbers in many other parts of the world including the United States [1]. Due to cancer related mortalities and the severe symptoms reported in cases with delayed diagnosis, several approaches for early detection and treatment modalities have been developed. For example, even though serum prostate-specific antigen (PSA) has been routinely used for screening individuals [2,3], recent studies proposed prostate-specific membrane antigen (PSMA) as a more precise and specific test for PCa screening and prognosis [4]. PSMA is an integral type II membrane glycoprotein that clusters upon activation at the cell surface, associates with lipid rafts, and is endocytosed [5,6]. In LNCaP cells, these events trigger a signaling cascade implicating the small GTPases RAS and RAC1 and the MAPKs p38 and ERK1/2 concomitant with an activation of NF-kappaB pathway and an increased expression of IL-6 and CCL5 genes [7]. Unlike serum PSA, PSMA is membrane-bound and not secreted and its examination requires a prostate biopsy, which is an invasive procedure.

In treatment protocols, androgen withdrawal was widely considered as a golden standard for patients suffering from metastatic PCa, giving favorable responses in a wide range of treated cases [8]. Nevertheless, in a short period of time following therapy, death resulting from androgen-independent progression may occur [8]. To achieve better survival rates in patients with metastatic PCa, efforts including chemotherapy and palliation using docetaxel and prednisone have been conducted [9]. Unfortunately, the therapeutic outcomes did not reach satisfactory levels highlighting the substantial need to identify and target the molecular mechanisms promoting PCa-resistance to the anti-cancer therapeutic treatments [8]. 

Resistance of cancer cells to the various treatments being used involves complex cellular processes; these include the induction of anti-apoptotic genes, adapting new signaling pathways [10,11,12,13], genetic instability coupled to clonal selection [14], and ligand-independent androgen receptor (AR) transactivation [8]. This complexity behind therapeutic resistance, that is mostly linked to metastatic cancers, is a major obstacle facing efficient cancer treatment and therefore constitutes a leading factor to mortality rates among treated patients [8].

In fact, recent research has adapted in-depth metabolic and biochemical investigations for studying the mechanism of occurrence and potential determinants of PCa metastasis [15,16]. These included various analytical platforms that facilitate the detection of certain metabolites or biological molecules known to promote tumor formation [16]. For instance, chromatography—including gas and liquid chromatography—coupled to mass spectrometry is widely used to assess the hydrophilic metabolite profiles in biological samples of tumors. In addition, assessment of redox homeostasis in terms of the GSH/GSSG (glutathione/oxidized glutathione) ratio is indicative of redox balance in cancer cells, which in turn, is crucial for tumor initiation and progression. Other compounds involved in cancer progression include serine, glycine, and one-carbon metabolites (SGOC) which contribute to the synthesis of glutathione and nucleotides, and maintain DNA methylation [16,17,18].

At the protein level, cancer cells including PCa cells are continuously subjected to proteotoxic stress, a condition that necessitates recruiting sophisticated cytoprotective mechanisms [19]. Heat shock proteins (HSPs) are highly conserved molecules acting as key regulators of proteostasis in both cancerous and non-cancerous cells. In cancerous cells, these molecules support the folding of cancer specific proteins and display anti-apoptotic properties [20,21,22]. Indeed, overexpression of HSPs has been useful in the diagnosis of many cancers including breast, endometrial, ovarian, esophageal, lung, colorectal, bone, urinary, and PCa [23]. Moreover, selected HSP members have been implicated in the prognosis of several cancers like renal, gastric, head and neck squamous cancers, ovarian and bladder cancers [23]. 

In PCa, specific chaperone members like HSP90, HSP70, HSP27, and clusterin are crucial regulators as they essentially contribute to AR folding and trafficking [19]. Certain studies report the implication of HSP60 in PCa development [8,24]. Some chaperones, like HSP27 and HSP60, serve as a prognostic tool to precisely stage the disease state [24]. This is of particular importance as heterogeneity in the patient’s pool with a range of prognosis is a key factor in high risk disease. For instance, some PCa patients have a lethal phenotype which may ultimately lead to death, while others may be cured early in the primary stage of tumor development [1].

In the present review, we present the various roles of heat shock proteins in PCa, the mechanism of their induction, anti-apoptotic function, and implication in cancer progression as well as chemoresistance. Additionally, we summarize the research efforts that have focused on targeting these chaperone molecules as a strategy to treat highly resistant PCa. 

## 2. Heat Shock Proteins: Functional and Regulatory Overview

Heat shock proteins (HSPs) are multifamily proteins that include highly conserved molecules known for their stress resistance. Historically, they were first identified after accidental thermal induction of insect cells [25]. Mammalian HSPs are grouped into six families according to their relative molecular mass: small heat shock proteins (sHSP) family, the HSP40 family, the HSP60 or chaperonin family, the HSP70 family, the HSP90 family, and the large HSP family [26]. Table 1 summarizes sample protein members from each family and their encoding genes in human.

Indeed, basal expression of HSPs occurs in normal eukaryotic cells, however, upon exposure to various physiological or stress insults such as hyperthermia, anoxia, ischemia, toxins, bacterial or viral infections, and UV light, their expression is dramatically increased, thus contributing to a key physiological response known as “heat shock response”(HSR) [44]. The HSR is universal in nearly all living organisms and wisely regulated in vertebrates by several transcription factors including heat shock factor 1 (HSF1), 2, 3, 4, and HSFY [45,46]. Structurally, all HSFs in vertebrates share the similarity of an N-terminal DNA binding domain and a C-terminal transactivation domain. HSF1 is considered as the master regulator of HSR [45,47,48,49] since it controls HSP gene expression by binding to the heat shock element (HSE), which is a specific DNA sequence located upstream of the HSP gene (Figure 1). Another important aspect in the HSR is that certain HSP members like HSC70, GRP78, MTP70, and HSP90β can be constitutively expressed, independent of the canonical HSR pathway [29,50,51].

Despite their important and numerous functions in cellular homeostasis, HSPs are principally appreciated for their “molecular chaperone” activities [52,53]. Classified as holdases and foldases, HSPs’ chaperone function is achieved through their binding to hydrophobic parts of the misfolded proteins to prevent their cytotoxic aggregation, assisting the folding of newly synthesized/nascent proteins to reach their functional 3D form, in addition to the refolding of misfolded proteins [54,55]. Furthermore, HSPs are implicated in a multitude of cellular processes including the regulation of cell signaling, protein assembly, trafficking, secretion, translocation, and degradation [56,57].

The structural architecture of HSPs has been previously reviewed [38,58,59]. Briefly, small HSPs that act independent of ATP are characterized by the existence of a conserved α-crystallin domain that is localized between N-terminal and C-terminal sequences [30]. On the other hand, the majority of large HSPs are functionally ATP-dependent and harbor ATP binding domains within their sequences to perform the ATPase activity [60,61]. A summarized diagram for the general structural architecture of sample important HSPs is presented in Figure 2.

## 3. HSPs in Cancer Cells

It is known that normal cells are equipped with distinct cytoprotective mechanisms to withstand various stress conditions. In this context, non-stressed cells express basal levels of HSPs which is enough to maintain proteome and cellular homeostasis. On the other hand, stressed cells express higher levels of HSPs enabling them to protect vital proteins from the proteotoxic agents. In cancer cells, the metabolic profile displays significant changes at the level of glycolysis, the tricarboxylic acid (TCA) cycle, oxidative phosphorylation, lipid metabolism, and amino acid metabolism [15]. In addition, other dramatic changes are involved in terms of dysregulated signaling pathways, increased oncogene levels, generation of high number of mutant proteins, and inhibition of tumor suppressor proteins [62]. Due to these serious alterations in cancer cell metabolism, the metabolic as well as proteomic loads are greatly intensified leading to a constant stressful situation that must be controlled to maintain cell functionality and survival. Therefore, it is not surprising that HSPs are key elements which are highly expressed and required in cancer cells. This high expression serves primarily to fold the oncoproteins accompanying tumorigenesis, maintain their stability, and counteract their potential aggregation or proteolytic degradation in the overcrowded tumor microenvironment (TME) [26].

## 4. HSPs in Prostate Cancer

One significant facet of HSPs’ roles in PCa cells arises from their ability to fold and support the trafficking of the androgen receptor (AR). This receptor is a steroid receptor transcriptional factor for testosterone and dihydrotestosterone which play a crucial role in PCa, particularly castration-resistant prostate cancer (CRPC) [19,63]. In prostate cells, AR is a client protein that interacts with set of chaperones including HSP90, HSP70, and HSP40 (Figure 3) as wells as co-chaperones like p23, FKBP-52, and α-SGT proteins [64,65]. Firstly, the cytosolic AR binds to HSP70 and HSP40. Upon binding of HSP90 and other co-chaperones, subsequent dissociation of HSP7 and HSP40 occurs. HSP90 maintains the proper AR conformation and safeguards its high affinity to its ligand, dihydrotestosterone [19,65]. Binding of AR to dihydrotestosterone results in its phosphorylation and dimerization. This is followed by the binding of HSP27 to the AR homodimer, supporting its trafficking to the nucleus where it can bind specific DNA sequences known as androgen response element (ARE). The binding of AR to ARE resides within the promoter sequences of certain genes and can hence control their regulation and transcriptional activity [19,66]. Therefore, HSPs represent ideal therapeutic targets in case of castration-resistant prostate cancer (CRPC) where the AR pathway is persistently active [19]. Moreover, inhibition of HSPs limits the chaperoning capability of other PCa oncoproteins and alters their associated signaling and transcriptional networks in CRPC cells [26]. For these reasons, HSPs have emerged as promising targets for therapeutic drugs in PCa.

### 4.1. HSP90

Having an approximate molecular weight of 90 kDa, the members of HSP90 family are localized into different cellular compartments including the cytoplasm, endoplasmic reticulum (ER), and mitochondria [38]. For instance, the cytoplasmic HSP90 isoforms are HSP90α and HSP90β while the ER resident and mitochondrial members include GRP94 and TRAP1, respectively [38,67]. These HSPs possess a characteristic ATPase domain and an overall similar structure that consists of three principal domains: the N-terminal domain (NTD), the middle domain (MD), and the C-terminal domain (CTD). The cytosolic HSP90 forms a homodimer whose binding activity is regulated by a subset of co-chaperones including AHA1 [68,69,70]. Furthermore, the HSP90 isoforms are subjected to diverse posttranslational modifications (PTMs) that are believed to modulate their function and interaction with the client proteins in normal as well as cancerous cells [71].

#### 4.1.1. Biological Functions of HSP90 Isoforms in Prostate Cancer

Due to their molecular chaperone activities, HSP90 members interact with diverse client proteins, estimated to be more than 200 clients, that are involved in copious cellular functions like cell growth, cell proliferation, and survival [72]. These critical proteins are necessary for cancer cells to develop and propagate. For instance, HSP90 chaperones apoptotic related proteins (Bcl-2, Apaf-1), tumor suppressor proteins (p53), mutant kinases, factors for angiogenesis (HIF1-α, VEGF) and steroid hormone receptors like androgen, progesterone, glucocorticoid receptors in addition to other proteins implicated in tissue invasion and metastasis such as (MMP2) [72] (Table 2). Of particular importance, HSP90 protects AR from potential degradation, thus maintaining its proper conformation, and sustains its high binding affinity to its ligand. Together with HSP27, HSP90 assists in AR nuclear trafficking. Furthermore, clients of HSP90 are associated with development and/or maintenance of CRPC such as protein kinase B (Akt), ERK-1/2, receptor tyrosine-protein kinase erbB-2 (ERBB2), proto-oncogene tyrosine-protein kinase Src (p60-Src), cyclin-dependent kinases (CDKs), and survivin.

It is of note that highly aggressive hormone-refractory PCa contains a prevalent magnitude of cells with neuroendocrine criteria. These prostatic neuroendocrine cells have been found to secrete many factors including parathyroid hormone-related protein (PTHrP), bombesin, and vascular endothelial growth factor (VEGF), which can influence PCa progression and responsiveness to anti-cancer drugs. Interestingly, HSP90/HSP70 chaperone complex functionality as well as HSP90 interacting affinity could be modulated in response to PTHrP. DaSilva and his colleagues have demonstrated that HSP90–AR interaction was lowered due to PTHrP induction and its effect favored stabilizing AR rather than its targeting to proteasomal-dependent degradation via the chaperone-associated ubiquitin ligase COOH terminus of Hsp70-interacting protein (CHIP) [73]. Due to this multi-faceted interference of HSP90 with the AR receptor pathway and the suppression/degradation of other oncogenic proteins, inhibition of HSP90 has been adapted as an attractive approach to treat mCRPC [19]. 

Recent reports have shown that the mitochondrial HSP90, TRAP1, is widely implicated in PCa development [74]. Increased expression levels of TRAP1 in PCa have been associated with malignant progression and metastasis of PCa [75]. Prostatic cancer cells have been shown to require TRAP1 for efficient mitochondrial respiration and withstand metabolic stress [76]. Furthermore, TRAP1 transgenic mice exhibited higher incidence of prostatic adenocarcinoma with elevated cell proliferation and reduced apoptosis while TRAP1 silencing hindered prostatic tumorigenesis in Pten^+/−^ mice [77].

GRP94, the endoplasmic reticulum HSP90 isoform, has also been involved in PCa metastasis. Stable GRP94 knockdown in the highly metastatic PCa cells, PC3-MM2, resulted in disruption of cell morphology and impaired the proper migration of the cells [78].

**Table 2 cancers-11-01194-t002:** Biological functions of HSPs in human prostate cancer.

HSP	Mechanism	Used cell Line/Model	Reference
Anti-apoptosis			
HSP90	Involved in several signaling and proliferative pathways via AR, ERBB2, Akt, c-RAF, survivin, EGFR, IGFR–1, STAT3, ERK, CDK-4, and CDK-6	PC3-MM2, LNCaP-LN3, VCaP, 22Rv1, DU145 and PC3, CWR22 and CWR22R xenografts	[79,80,81,82,83,84,85,86]
HSP70	Suppresses the pro-apoptotic protein MST-1, resulting in cisplatin resistance	DU145	[87]
	Stimulates overexpression of Bcl2-L-3, BCL2, and Bcl2-L-1, thus making PCa cells resistant to ionizing radiation and etoposide.	PC3 and LNCaP	[88]
HSP27	Hampers STAT3-regulated apoptosis, leading to resistance to androgen withdrawal	LNCaP	[89]
	Increases the expression of eIF4E, making PCA cells refractory to androgen withdrawal and paclitaxel	LNCaP	[90]
	Upregulates TCTP, which diminishes docetaxel-mediated apoptosis in LNCaP cells	LNCaP	[91]
	Promotes IGF-1-induced phosphorylation of ERK, Akt and RPS6KA, thus inactivating the BAD-14-3-3 protein complex and inhibiting apoptosis	PC3	[92]
	Hinders Fas-mediated apoptosis by allowing PEA–15 to bind FADD in an Akt-dependent mechanism	LNCaP	[93]
Clusterin	Counteracts Bcl2–L–4–mediated caspase activation, resulting in apoptosis inhibition in PCa cells treated with camptothecin and etoposide	PC3 and DU145	[94]
	Inhibits apoptosis in rat prostatic cells treated with actinomycin D through phosphorylation of Akt, subsequent phosphorylation of BAD and reduced cytochrome c release	MLL Dunning rat prostatic adenocarcinoma cell line	[95]
	After stimulation by Akt, clusterin causes resistance to docetaxel	DU145 and PC3	[96]
	Its overexpression upon enzalutamide treatment of PCa cells confers resistance and inhibition of apoptosis. The process occurs through the RPS6KA–YB–1 signaling pathway and involves clusterin mediated activation of Akt and MAPK	LNCaP	[97]
	Inhibits paclitaxel-induced apoptosis after GRP78 mediated translocation to other cellular compartments including cytosol and mitochondria	LNCaP	[98]
AR, trafficking, stability, and transcription regulation			
HSP90	Protects against AR degradation	LNCaP, PC3-MM2, LNCaP-LN3, VCaP, 22Rv1, DU145 and PC3, CWR22 and CWR22R xenografts	[79,80,81,84,86,99]
	Preserves high-affinity ligand-binding conformation of AR	LNCaP, Yeast cells	[65,99]
	Aids nuclear trafficking of AR, thus facilitating transcription of AR-regulated genes such as PSA and expansion of castration resistance	LNCaP cells	[100]
HSP70	Facilitates binding of BAG-1 to N-terminus of AR thus stimulating enhanced transcription of AR-regulated genes like KLK3	PC3. LNCaP, 22Rv1 and CWR22R xenografts	[101]
Invasion and metastasis			
HSP90	Together with its client proteins, HSP90 is implicated in enhanced lymph node metastases	PC3LN3 orthotopic lymph node mPCa model	[102]
	Activates NF-κB and p60-Src leading to RANKL-induced osteoclast differentiation	LNCaP xenograft model	[86]
	Enhances PCa cells motility through ERK and MMP-2-MMP-9	DU145 and LNCaP, ARCaP	[103]
HSP70	Besides HSP90, HSP70 is involved in WASF3 metastasis-promoting protein stability and activity	PC3 cells	[104]
HSP27	Implicated in TGF-β-mediated MMP–2 activation and invasion	PC3 cells	[105]
	Augments EMT via IL-6-STAT3-Twist signaling resulting in increased cell migration and invasion and metastases	PC3M model	[106]
Clusterin	Implicated in TGF-β-mediated invasion	PC-3	[107]
	Initial signals coming from Twist1 and TGF-β activates clusterin to promote EMT and increase metastasis	PC3M model	[108]
Angiogenesis			
HSP90	Protects HIF-1α from proteasomal degradation, leading to increased expression of VEGF and angiogenesis	DU145, PC3 and LNCaP	[109,110]
Clusterin	IL-24 reduces secretory clusterin levels, thus diminishing angiogenesis	DU145 xenografts	[111]
Cell cycle regulation			
HSP90	Modulation of HSP90 interactors like AR, ERBB2, and Akt is associated with enforcement of G1 cell-cycle checkpoint	DU145 and LNCaP	[84]
	Modulation of HSP90 client proteins including AR, ERBB2, Akt, c-RAF, and CDK-4 is critical for enforcement of G2–M cell-cycle checkpoint	PC3 and LNCaP	[82]
HSP27	Induces TCTP overexpression resulting in maintaining prostatic cells in S-phase of the cell cycle	PC3 and LNCaP	[91]
	Induces PEA-15 phosphorylation, thus hindering its association to ERK. This allows for ERK nuclear translocation and promoting the cells to stay in S-phase of the cell cycle	LNCaP	[93]

#### 4.1.2. Targeting HSP90 in Prostate Cancer

Due to their involvement in PCa, all HSP90 isoforms seem to be attractive targets in PCa. In this regard, many inhibitors of HSP90 have been developed with the majority of inhibitors having cross targeting properties of other HSP90 isoforms. However, few HSP90 inhibitors that have been used for treatment of other cancers are selective and target specific isoforms such as the newly discovered compound 54 [112] and Br-BnIm [113] for GRP94, in addition to shepherdin and SMTIN-P01 for TRAP1 [74]. In the current review, we focus on the cytosolic HSP90 inhibitors that have been tested in PCa studies. 

The fact that HSP90 functions through its ATPase activity encouraged researchers to target this chaperone at its ATP binding pocket which is located within its N-terminal domain [114]. Several HSP90 inhibitors have been currently identified and used in preclinical and clinical trials. Even though the exact anti-tumor action of such molecules is yet unclear [72]. A significant part of the underlying molecular mechanism of HSP90 inhibitors relies on the HSP90 complex conformation and interference with its folding cycle. Interestingly, tumor-derived HSP90 has 20-to-200-fold higher affinity to the small molecule inhibitors as compared to HSP90 extracted from normal cells [115]. 

#### Small Molecule Inhibitors of HSP90

Generally, most of the HSP90 inhibitors belonging to this category are designed to mimic the structure of ATP that is supposed to bind the N-terminal domain of HSP90, thus blocking ATP binding [116,117]. This results in interruption of the interaction of HSP90 with its client proteins and consequently their potential proteasomal degradation. The most famous early discovered HSP90 inhibitors include geldanamycin (GA), a benzaquinoid ansamycin antibiotic and radicicol (RD) which is a macrocyclic antifungal antibiotic. Despite their powerful effects, these molecules had poor solubility and high toxicity at therapeutic doses besides their instability [114]. Recent advances and technologies in drug design allowed the development of a new generation of synthetic compounds, some of which have been used to treat PCa and others still under clinical trials (www.clinicaltrials.gov) [118]. Another group of inhibitors has been identified to function without targeting the HSP90 N-domain; these include novobiocin and its derivatives, gedunin and celastrol, that bind the HSP90 C-terminal domain hindering its interaction with co-chaperones and client proteins [119,120,121]. Moreover, certain drugs like sulphoxythiocarbamates target the middle domain by modifying cysteine residues resulting in conformational changes that ultimately inhibit client binding [122]. In the following sections, we focus on N- and C-terminal domain inhibitors of HSP90. Additionally, Figure 4 provides a summarized overview about HSP90 inhibitors commonly used for cancer treatment.

#### HSP90 Inhibitors in Pre-clinical Prostate Cancer Studies

N-terminal domain inhibitors

One of the first-generation inhibitors targeting the N-terminal domain of HSP90 is tanespimycin (17-N-Allylamino-17 demethoxygeldanamycin, 17-AAG) which enhanced the degradation of AR in PCa cell lines and in mouse xenograft models [123]. Compared to the control group, LuCaP35 xenograft tumors that were treated with 17-AAG showed delay in transition to CRPC due to the blocked nuclear translocation of AR [100,124]. 

On the other hand, new generation inhibitors included ganetespib, NVP-AUY922, SNX-2112, and SNX-5422 that showed distinctive anti-cancer efficacy in both cell culture as well as pre-clinical trials. For instance, ganetespib has been used in many clinical trials as a monotherapy or in combination with docetaxel in NSCLC adenocarcinoma [125]. Interestingly, ganetespib induced apoptosis in androgen sensitive as well as resistant cell lines including LNCaP, VCaP, Du145, and PC3 [81]. Moreover, weekly treatment of PC3 and 22Rv1 xenografts in mice with ganetespib caused delay in growth and downregulation of AR ligand-incompetent transcript variant V7 [81]. Notably, in CRPC many AR variants do not rely on Hsp90 complex formation for ligand binding, and can consequently be referred to as non-HSP90 clients. Interestingly, however, some of these variants remain sensitive to ganetespib either alone or in combination [126].

Another new-generation synthetic small molecule inhibitor of Hsp90 is NVP-AUY922. This HSP90 inhibitor exhibited anti-tumor activities in several cell lines and mice xenografts [102]. Derivatives including NVP-AUY922 and NVP-HSP90 displayed similar effects to 17-AAG in terms of protein client degradation and HSP70 upregulation in cultured primary prostate tumors [82]; Nevertheless, both compounds exhibited greater apoptotic and anti-proliferative effects than 17-AGG [82,125].

SNX-2112 (PF-04928473) and its prodrug SNX-5422 (PF-04929113) are new generation anti-HSP90 drugs that showed remarkable apoptotic effects in LNCaP cells [86]. Moreover, they resulted in a substantial delay in the growth of LNCaP xenografts in mice. Furthermore, in contrast to 17-AAG, SNX-5422 could inhibit osteoclastogensis and bone metastasis, encouraging its further clinical applications [86,118].

C-terminal domain inhibitors

These drugs are able to attach to the C-terminal of HSP90 and suppress its activity without stimulating HSR that likely counteracts cellular apoptosis [127]. KU174 is a second-generation novobiocin analogue that causes degradation of HSP90 in PC3-MM2 and LNCaPLN3 PCa cells. Notably, treatment with KU174 has been suggested to preferentially target Hsp90β-containing complexes resulting in many consequences. For instance, it caused reduction in cell viability in case of PC3-MM2 cultures and decreased tumor size in rats with PC3-MM2 xenografts [80]. Based on their research in this area, Vielhauer and Blagg suggested that various C- and N-terminal domain inhibitors of HSP90 can target specific protein clients [85,118]. Another novobiocin analogue that acts without HSR induction is F-4. This compound has higher potency, in terms of apoptosis induction and suppression of proliferation in PC3 and LNCaP cells, as compared to 17-AAG when given at equivalent concentrations [85]. Table 3 presents the key drugs that target HSPs for PCa treatment.

#### Clinical Assessment of HSP90 Inhibitors in CRPC

As previously discussed, HSP90 inhibitors showed promising outcomes in preclinical studies on PCa. However, in real clinical applications there were considerable limitations that faced primary trials concerning toxicity and bioavailability. Here, we summarize some clinical trials and describe the restrictions that aroused upon the application of some first-generation HSP90 inhibitors.

New-generation HSP90 inhibitors like the resorcinol derivative, ganetespib has been included in a Phase II trial of mCRPC in patients that were initially treated with docetaxel (NCT01270880, www.clinicaltrials.gov/). Interestingly, the same combination of drugs, ganetespib and docetaxel, has been used in a Phase III trial of advanced non-small cell lung cancer and showed better safety profiles when compared to HSP90 first generation inhibitors [125]. Furthermore, ganetespib is currently being used in many clinical trials for other types of cancers including hepatic and breast cancers, malignant peripheral nerve sheath, and ovarian and fallopian tube cancers [118]. AT13387 is a second-generation HSP90 inhibitor that has been used in another clinical trial (NCT01685268, www.clinicaltrials.gov/) as monotherapy or in combination with abiraterone acetate and prednisone. CRPC patients resistant to abiraterone and prednisone were the targets of this trial. Among the clinical purposes of such studies were safety and dosing schedule, monitoring PSA levels and tumor volume. Detailed information about the latest, up-to-date studies concerning clinical trials of PCa, as well as other cancer types are available through the resource www.clinicaltrials.gov/.

#### HSP90 Inhibitors in Combination Therapies for Prostate Cancer

To combat severe malignancies, co-targeting of several oncoproteins has been envisioned as an effective treatment strategy due to the increased efficacy and synergistic action [138,139] Indeed, inhibition of HSP90 sensitizes PCa cells to both radiation therapy and chemotherapies [140,141]. For instance, 17 AAG showed greater efficacy after being combined with either proteasome or HSR inhibitors in PC3 cells [142]. Moreover, combination of GA and TRAIL (an activator of extrinsic apoptosis pathway) could suppress TRAIL-resistant LNCaP cells [143]. Furthermore, co-targeting of multiple molecules including mTOR, ERK, and HSP90 using rapamycin, CI-1040, and 17-AAG, respectively resulted in profound apoptotic effects and overexpression of the tumor suppressor protein, E- cadherin [144].

NF-κB is a transcription factor that has been involved in the transcription of many genes including CXCL8 and possibly HSP90 [145]. In PCa, high expression levels of the CXCR1/2 ligand CXCL8 (IL-8) and its receptors CXCR1 and two have been associated with the progression to CRPC. Combination therapies including GA and AZ10397767, a CXCR2 antagonist, or GA and BAY11-7082, a NF-κB inhibitor, reduced the viability of PC3 cells [145]. Strikingly, the same therapeutic combination did not affect the viability of DU145 cells suggesting that metastatic PC3 cells, contrary to DU145 cells, are highly dependent on CXCL8 signaling [118]. Other contradicting studies report that HSP90 inhibitors such as ganetespib are cytotoxic in PCa cell lines irrespective of AR sensitivity or metastatic potential [81] (Table 3). These data reflect variations among first-generation and new- generation Hsp90 inhibitors.

In other studies, suppression of Wee1 kinase enhanced the cytotoxic potential of HSP90 inhibitors including 17-AAG, ganetespib, SNX2112, and RD [146,147]. The tyrosine kinase Wee1 is an HSP90 client that controls G2/M cell cycle progression and phosphorylates Y38 of human HSP90 [148]. Interestingly, the missense mutation of a conserved tyrosine residue in Wee1 to Y38F, a non-phosphorylated form, disrupted the HSP90 chaperoning machinery involving v-Src and HSF1 [147]. In addition, inhibition of Wee1 in PC3 xenograft model led to hypersensitization of the cells to Hsp90 inhibitors and evoked the intrinsic apoptotic pathway [146].

It is worth noting that dual targeting of HSP90 and HSP27 or HSP90 and clusterin, which are also molecular chaperones, has been adapted as an effective anti-tumor treatment strategy as will be explained in the following sections. Collectively, these interesting findings strongly support the beneficial effect of combination therapies and encourage further combination approaches to defeat highly resistant prostate tumors.

#### 4.1.3. Extracellular HSP90 (eHSP90) Is an Attractive Target in Metastatic Prostate Tumors

Released forms of HSP90 have been reported in normal as well as cancerous cells, suggesting a plethora of signaling and immunological roles for these secretory proteins [149,150,151,152]. In fact, a primary key step for the transformation of cells into a metastatic state includes the disruption of E-cadherin [153]. This process is critical for early epithelial to mesenchymal transition (EMT). Recent reports strongly suggest eHSP90 as a chef regulator of PCa metastasis [103]. As evidence, overexpression of eHSP90 led to downregulation of E-cadherin and increased levels of matrix metalloproteinases via Slug stimulation [103]. Other supporting proof for its metastatic roles include the high expression levels of eHSP90 in PCa cell lines and suppression of metastatic events upon treatment of ARCaPE cells with non-permeable GA [154]. Moreover, secretory exosomes of LNCaP cells that were cultured in hypoxic conditions contained eHSP90 that could elicit an invasive phenotype in PC3 cells [155]. Furthermore, hyperacetylated eHSP90 was found to restrain tumor cell invasion in T47D breast cancer cells [156]. These findings pinpoint PTMs of eHSP90 as promising target in mCRPC and reflect various modalities through which eHSP90 can be targeted [118].

#### 4.1.4. HSP90 Regulation as Potential Target in Prostate Cancer

Two regulatory mechanisms of HSP90 have been considered as attractive targets in PCa treatment, namely posttranslational modification and co-chaperones. Generally, posttranslational modifications play a key role in modulation of chaperone functions [157]. In cancer, modification of several oncoproteins and chaperones have been associated with therapeutic resistance or sensitivity, cancer progression, and metastasis [117]. Examples of these include acetylation, phosphorylation, oxidation, and S-nitrosylation [19]. 

#### Posttranslational Modification of HSP90

Modification of HSP90 by hyperacetylation was observed among several consequences following histone deacetylase 6 (HDAC6) inhibition. As a result, the ATP-chaperoning machinery of HSP90 was disrupted, client proteins were degraded and HSP90 had a higher affinity towards 17-AAG [158]. The HDAC inhibitor PXD101 was shown to degrade AR and induce apoptosis in LNCaP, C81, and 22Rv1 cell lines. In addition, it caused reduction of tumor size in 22Rv1 mice xenografts [159]. Furthermore, Romidepsin, a pan-HDAC inhibitor has been used in phase II clinical trials of PCa. This drug acts through increased acetylation of histones as well as HSP90 and diminishing the AR function [160]. Despite the positive response of some patients, as revealed by reduction of PSA levels, side effects including nausea, fatigue, and vomiting caused cessation of the trial for several cases [160]. Collectively, these findings suggest HDAC inhibition as a therapeutic approach to prevent progression of PCa to CRPC. However, safety issues have to be taken into consideration for evaluating the overall therapeutic approach [118].

#### Co-Chaperones

Similar to posttranslational modification of HSP90, targeting co-chaperones represents an indirect way in altering the chaperone machinery to combat PCa. Many of HSP90 co-chaperones like Cdc37, P23, FKBP52, and SGTA have been involved in the pathogenesis of PCa [118] (Table 4).

### 4.2. HSP70

The family of HSP70 proteins represents one of the most studied HSPs with respect to cellular stress. According to Brocchieri et al., the human genome codes for 13 HSPs belonging to the HSP70 family [171,172]. These proteins share a highly conserved structure consisting of an N-terminal domain (about 45 kDa) with ATPase activity and a C-terminal substrate binding domain (about 25 kDa) (Figure 2) [59,173]. Members within the HSP70 family are grouped according to their expression behavior into either inducible or constitutive proteins. The most strongly inducible members are HSPA1A, HSPA1B, and HSPA6 [36,173]. Example of housekeeping or constitutively expressed HSP70 proteins include HSC70 (HSP73 or HSPA8) [174]. Under non-stressed states, HSP70 proteins assist in the folding of nascent polypeptides and promote the assembly and transport of proteins across cellular membranes [59,175]. In stressful situations like cancers, HSP70 expression is significantly increased, performing multiple cellular functions to sustain cell viability [176].

#### 4.2.1. Biological Functions of HSP70 in Prostate Cancer

High expression of HSP70 is believed to be essential for the survival of many cancer cells [20,177]. In PCa, the inducible HSPA1A (HSP70.1 or commonly known as HSP70) is involved in the early formation of AR-chaperone complex in an ATP requiring process. HSP40, another HSP belonging to the HSP40 family which possesses a characteristic J-domain, mediates the binding of AR to HSP70 and stimulates the ATPase activity of HSP70 chaperone. Subsequent binding of HSP90 to the complex AR-HSP complex leads to the dissociation of HSP70 and HSP40 as previously described [19,64]. The collaborative action of the chaperone complex HSP70/HSP40/HSP90 ensures the maintenance of AR in a high affinity ligand binding state [178]. Together with HSP27, HSP70 has been believed to be crucial in de novo angiogenesis, invasion, and metastasis. 

Importantly, the ER resident HSP70 isoform, GRP78 or BiP, has been strongly associated with castration resistance prostate cancer (CRPC) both in vivo and in vitro [179]. For instance, patients with greater expression of GRP78 had higher risk of PCa recurrence and lower survival rate in patients with no prior hormonal manipulation [179,180]. Moreover, GRP78 levels were higher in the castration resistant LNCaP-derived cell line, C42B and in LNCaP cells maintained in androgen-deficient conditions as compared to LNCaP cells grown in androgen-rich media [179]. Furthermore, GRP78 knockdown sensitized PCa cells to the anticancer agent Shikonin [181]. Recent reports have shown that prostatic tumors metastasizing to the bone and bone marrow space transmit their prosurvival signals via direct adhesive interactions with the surrounding stromal elements, thus resisting cytotoxic effects of drug treatment [182]. In this regard, GRP78 expression has been linked to proper functioning of N-cadherin (N-cad), a key player in the adhesive interactions of metastatic PCa with the bone microenvironment. GRP78 knockdown PCa cells also exhibited lower expression levels of E-cadherin (E-cad), altered morphology, and impaired adhesion to osteoblasts [182].

Lately, several reports hypothesize tumor cells to be addicted to chaperones, highlighting the great dependence of oncoproteins on HSP70 and other chaperones for their folding and proper function [183,184]. In PCa cells, HSP70 inhibits the mammalian STE20-like protein kinase 1(MST-1) mediated apoptosis [87] and promotes the overexpression of anti-apoptotic proteins such as myeloid leukemia cell differentiation protein-1 (Mcl-1 or Bcl2-L-3), apoptosis regulator Bcl-2 (BCL2), and Bcl-2-like protein 1 (Bcl2- L-1 or Bcl-X(L) [88]. Additionally, HSP70 was found to potentiate the stability of WASF3, thus increasing metastasis and invasion [104]. Critically, overexpression of HSP70 renders CRPC cells insensitive to radiation therapy and increases their resistance to certain drugs like etoposide [88], doxorubicin [185], and cisplatin [87].

#### 4.2.2. Targeting HSP70 in Prostate Cancer

Like HSP90 inhibition, suppression of HSP70 that is highly needed and substantially expressed in cancer cells, is an attractive approach for PCa treatment. Short interfering RNA (siRNA) of HSP70 is an adapted strategy that augmented the pro-apoptotic activity of some anti-tumor drugs like etoposide in PCa cells [88]. Moreover, HSP70-siRNA lessened the clonogenic survival of cancer cells that were treated by ionizing radiation, cisplatin, vinblastine, or paclitaxel [186]. Nevertheless, translation of these findings in vivo remains challenging [19]: relative to HSP90 and HSP27, fewer studies have investigated the effects of HSP70 inhibition in PCa cells. The following section describes MKT-077, which is the only HSP70 inhibitor used in a PCa clinical trial, in addition to other inhibitors that have been used in other cancer types, and may be prospective candidates for PCa treatment [19].

#### *N*-terminal Domain or ATPase Domain Inhibitors

ATPase inhibitors of HSP70 have been designed based on their analogy to their correspondents of HSP90 inhibitors. Initial designing trials relied on investigating the X-ray structure of commercially available ATP analogues [187]. However, drug specificity and potency aspects remained doubtful.

VER-155008 is an adenosine derivative that inhibits the HSP70 chaperone function by binding its ATPase domain. This compound was used as monotherapy in experiments including breast cancer and colon cancer cell lines where it could initiate either caspase-dependent or caspase-independent apoptosis [188]. Interestingly, VER-155008 increased the apoptotic potency of small molecule Hsp90 inhibitor in HCT116 cells [188]. Other HSP70 inhibitors included azure C, methylene blue, and myricetin which are subject to investigation to ensure their specificity towards inducible HSP70 members [189].

MKT-077 is another cationic rhodacyanine dye which performs its anticancer action by targeting the ABD of mitochondrial HSP70. Despite its anti-tumor action in PCa DU145 xenografts, clinical doses of MKT-077 in treated patients caused severe renal pathologies in terms of renal magnesium wasting, leading to halting its clinical trials [190].

Other ATPase inhibitors comprise NSC 630668 and its newer generation MAL3-101. In addition to its ATPase inhibition activity, MAL3-101 was found to prevent the proliferation of SK-BR- 3 cancer cells [191].

#### *C*-Terminal Domain or Peptide Binding Domain (PBD)

HSP70 has been shown to impede a caspase- independent cell death through its binding to apoptosis inducing factor (AIF) [192]. Decoy targets for HSP70 have been employed as a recent approach to inhibit HSP70 mediated apoptosis. These designed inhibitors include peptides that are derived from apoptosis inducing factor (AIF) and most of them carry a specific amino acid sequence starting from amino acid number 150 to amino acid 228 [72,193]. For instance, ADD70 (AIF derived decoy for HSP70) could bind and suppress HSP70 function. This has been translated by its capability to cause reduction in tumor size and enhance chemotherapeutic sensitivity in rats including colon cancer cell model and mouse model of melanoma (B16F10) [194]. 

2-phenylethynesulfonamide (PES) is a small molecule inhibitor that associates with the C-terminal domain of HSP70. This association interferes with the HSP70 chaperone function leading to increased protein aggregation, lysosomal membrane instability, and ultimately autophagy [195]. Other reports demonstrate that PES induce apoptosis via a caspase dependent fashion [196]. In all cases, PES induced death has been reported to be cell type dependent [72].

#### Inhibitors of HSP70 Co-Chaperones

Other elegant ways to inhibit HSP70 is to target its defined co-chaperones. Interruption of HSP70-co-chaperones interaction limits the ability of HSP70 to carry out its proposed function [72]. Chemicals disrupting Hop/HSP70 interaction such as pyrimidotriazinediones exhibited cytotoxic effects in WST-1 cells [197]. Moreover, depletion of co-chaperone Hip impaired the HSP70 activity in neural cells and suppressed its protein homeostasis functions [198].

#### HSP70 Inhibitors in Combination Therapies

In many cancer therapies, overexpression of HSP70 has been reported to occur following treatment with HSP90 inhibitors. This finding was observed both in vivo and in vitro in cancers like leukemia and glioma [199,200,201]. In fact, several studies support this hypothesis: suppression of HSP70 by siRNA significantly sensitized cells to the HSP90 inhibitor, 17-AAG [202]. Similar results were obtained in HCT116 colon carcinoma cells where the HSP70 inhibitor VER-155008 enhanced the apoptotic inducing effect of 17- AAG [188]. In line with previous reports, ADD70, an HSP70 inhibitor, maximized the activity of 17-AAG in colon cancer cells [194]. Overall, treatment regimens including co-inhibition of HSP70 with HSP90 are likely to be among the future perspectives regarding PCa therapy.

### 4.3. HSP60 

HSP60, also known as chaperonin, is a member of HSP60 family which is localized mainly in the mitochondria [35,203]. This mitochondrial chaperonin is constitutively expressed in mammalian cells and acts together with its 10 kDa co-chaperone protein, known as HSP10, to fold proteins freshly transported into the mitochondria [34]. As an ATP dependent chaperone, human HSP60 oligomerization is enhanced at high ATP levels while lower ATP concentrations favor dissociation of HSP60 into monomers [204]. Unlike the bacterial homologue that is found in tetradecameric conformation, human HSP exists as homo-oligomer of seven subunits [205]. Interestingly, human HSP60 has three cysteine residues (Cys237, Cys442, Cys447) that do not exist in GroEL and represent potential nucleophilic binding sites for electrophilic HSP60-binding agents [206]. Interestingly, both HSP60 and HSP10 have been reported to exhibit other non-chaperone functions in various cellular locations including the cytosol, nucleus, intracellular vesicles, outer mitochondrial surface, and extracellular environment including blood circulation [34].

#### 4.3.1. Biological Functions of HSP60 in Prostate Cancer

Overexpression of HSP60 has been reported in several malignancies including breast [207,208], colon [208,209], lung [210], ovarian [211], and PCa [212,213]. Increased protein levels of HSP60 and HSP10 have been detected in both early and advanced PCa [214]. Moreover, immunohistochemical studies of PCa tissues revealed high HSP60 expression all over the prostate epithelium in contrast to limited expression of the chaperonin in the basal cell layer of normal prostate glands [215]. These findings, together with those obtained by Glaessgen et al., 2008 highly suggested HSP60 as a biochemical marker for PCa progression as well as its recurrence after radical prostatectomy (RP) [24]. Overall, the high expression of HSP60—especially in the early phase of carcinogenesis in addition to its potential implication in cancer development—opens avenues towards targeting this HSP in future cancer treatment protocols [8].

#### 4.3.2. HSP60 as Potential Target in Cancer Therapy

Since HSP60 has been linked to many cancers, there was a great interest to target this chaperone with small molecule inhibitors. Substances targeting HSP60 have been classified into derivatives of natural products and synthetic substances [34]. Mizoribine, an imidazole nucleoside antibiotic that is derived from *Eupenicillium brefeldianum*, has been reported as the first HSP60 inhibitor [216]. Even though Mizoribine lacks anti-microbial activities, it has been clinically used following kidney transplants due to its powerful immunosuppressive characteristics [216]. Its association to HSP60 results in suppression of its ATPase activity and disruption of the formation of HSP60-HSP10 complex [217]. Unfortunately, mM concentrations of Mizoribine are required to inhibit the HSP60 activity, whereas in clinical trials, the highest concentration that could be implemented was only 30 μM [218]. Therefore, more efforts are needed to establish a protocol with applicable effective therapeutic concentrations.

Epolactaene is another natural product originating from the fungal strain *Penicillium sp*. BM 1689-P and known to inhibit HSP60 [219,220,221]. Moreover, its tret-butyl ester ETB has been used as the active form of the compound, even though its exact mechanism of action is still unclear. Similar to Mizoribine, binding of ETB to HSP60 leads to inhibition of HSP60/HSP10 complex chaperoning activity [220].

More recent natural HSP60 inhibitors include myrtucommulone A (MC). Pull-down assays showed strong interaction between MC and HSP60 [222]. Importantly, this HSP60 inhibitor can be used in nanomolar concentrations and has displayed encouraging mitochondrial inhibiting effects in human leukemia cells. These included disruption of mitochondrial membrane potential (∆ψm), diminishing mitochondrial viability and prevention of mitochondrial ATP synthesis [223].

On the other hand, synthetic HSP60 inhibitors include o-carboranylphenoxyacetanilide which showed high affinity to HSP60 compared to other HSPs like HSP70 or HSP90 [224]. Additionally, gold (III) porphyrin complexes have been found to inhibit HSP60 and displayed anti-cancer effects in many cancer cell lines [225]. However, clinical application of HSP60 inhibitors in PCa field is still lacking, probably due to the deficient understanding of the molecular mechanisms underlying some compounds or due to toxic limitations.

### 4.4. HSP27

In human, HSP27 belongs to the family of small HSPs comprising 11 members that are characterized by a distinct conserved signature known as ‘crystalline domain’. This conserved amino acid sequence is located between variable N- and C-terminal domains [36,226] (Figure 2). Members of this family have a size range from 12–43 kDa and act in an ATP independent manner [58]. Importantly, these molecules perform their chaperone function by binding to aberrantly-folded proteins or misfolded and hydrophobic peptides to prevent their aggregation inside the cell [227]. Indeed, the most frequently studied members in this family are HSPB1 (HSP27), HSPB4 (αA crystallin), and HSPB5 (αB crystalline or CRYAB) [36]. Though high expression of sHSPs has been reported in many cancer types, several studies have focused on HSP27 or HSPB1 which is considered as a central sHSP in relation to PCa [228].

Similar to most small HSPs, HSP27 is able to form homo-or hetero-oligomeric complexes and is subjected to posttranslational modification. HSP27 can form oligomers having approximate molecular masses of up to 800–1000 kDa. Phosphorylation of three serine residues (Ser15, Ser78, and Ser82), located within the N-terminal domain, is a key posttranslational modification that affects HSP27 oligomerization status and chaperone function [28,30,229]. Other phosphorylation sites include Thr143 that is located in the α-crystallin domain [230]. Notably, MK2/3 (MAPKAP2/3) kinase dependent phosphorylation causes large HSP27 oligomers to dissociate forming small oligomers or even monomers. These structural changes are critical for HSP27 interaction and function [231]. Generally, large HSP27 oligomers which function independently of ATP are thought to bind misfolded proteins and prevent their aggregation, while small oligomers facilitate ubiquitination and degradation of client proteins [72,232].

#### 4.4.1. Biological Functions of HSP27 in Prostate Cancer

HSP27 performs multi-cellular functions in PCa cells (Table 2). As previously demonstrated, HSP27 contributes to AR nuclear trafficking through helping its palmitoylation, binding to the ARE, and performing its transcriptional activity [233]. Together with STAT3, eIF4E, and TCTP, it confers resistance to apoptotic induction following androgen withdrawal and taxane cytotoxins [89,90,91]. Strikingly, the anti-apoptotic actions of HSP27 are seriously impaired upon suppression of STAT3, eIF4E, or TCTP highlighting their roles in HSP27-induced cytoprotection [19]. It seems that HSP27 supports ERK and Akt phosphorylation in an IGF-I dependent manner, leading to BAD/14-3-3 complex stability [92]. HSP27 has also been reported to control the 15 kDa phosphoprotein enriched activity (PEA-15) leading to increased cell proliferation and concomitant inhibition of Fas-induced apoptosis [19,93]. Moreover, overexpression of HSP27 in LNCaP cells elicited PEA-15 phosphorylation at Ser116 in Akt mediated process. As a consequence, phosphorylated PEA-15 no longer sequestrates ERK allowing its translocation to the nucleus. Meanwhile, Phospho-PEA attaches to Fas-associating death domain-containing protein (FADD) leading to apoptosis suppression. Collectively, these results suggest dual concurrent proliferative and anti-apoptotic activities of HSP27 in PCa cells [93] which has promising clinical implications: patients with low phosphatase and tensin homolog on chromosome 10 (PTEN) expression and subsequent high Akt levels are expected to respond positively to HSP27 inhibitors [19].

Interestingly, HSP27 has been reported to be associated with therapeutic resistance of neuroendocrine PCa cells (NEPCa). These chemoresistant NEPCa cells may arise following androgen deprivation therapy with anti-androgens and, as previously mentioned in the HSP90 section, can even confer drug resistance to their neighboring PCa cells. Strikingly, PTHrP that is frequently secreted from NEPCa cells has been found to modulate p38/MAPK/Hsp27 signaling in the neighboring PCa cells. Activation of these signaling molecules impacts the activity of AR through increasing its nuclear translocation and upregulation of p21 which ultimately leads to chemoresistance to many anti-cancer drugs, including docetaxel [234]. 

Additionally, HSP27 promotes the development of malignant criteria through translationally controlled tumor protein (TCTP) as tested in PCa cells [91]. For instance, it stimulates the activity of MMP-2 via TGF-PCaβ pathway, promoting cell migration and invasiveness [105]. Furthermore, it modulates the epithelial–mesenchymal transition (EMT) in a way that activates IL-6–STAT3–Twist signaling [106].

#### 4.4.2. Prognostic Value of HSP27 in Prostate Cancer

Though being expressed in various cancer types, many reports appreciate HSP27 as a substantial marker in PCa. Increased expression of HSP27 in PCa cells indicates aggressive nature of the tumor, reflecting its metastatic tendency in addition to poor clinical outcome [235,236]. Recent proteomic studies suggest HSP27, together with ALDH6A1 and prohibitin, as a triple biomarker to monitor early as well as late stages of prostatic tumors [237]. Interestingly, in patients with localized PCa, high expression level of HSP27 has been found indicative of bad prognosis and high risk to develop CRPC even in cases which lack the canonical ETS gene rearrangement [89,92,238]. Remarkably, persistent overexpression of HSP27 has been reported in all CRPC cases and likewise, comparable elevated HSP27 levels occur following treatment with androgen ablation [89,92,236,239].

#### 4.4.3. Targeting HSP27 in Prostate Cancer

Unlike HSP90 and HSP70, HSP27 is not a suitable target to small-molecule inhibition due to the absence of an ATP binding domain in its structure. Other targeting mechanisms, such as siRNA, have been shown to suppress HSP27 anti-apoptotic capability and could limit its role regarding cellular proliferation [89,92,93,236,240,241]. These effects have been employed in treatment regimens where inhibition of HSP27 rendered PCa cells more responsive to paclitaxel [236]. Antisense nucleotides have been also used to hinder the expression of HSP27 in PCa cell by binding to its mRNA and preventing its translation [139,242]. For instance, OGX-427 is an antisense oligonucleotide that acts on HSP27, promoting the degradation of AR in LNCaP cells, leading to increased cell death. Moreover, OGX-427 treatment of mice containing LNCaP xenografts resulted in apoptosis, lowering the metastatic potential and the reduction of serum PSA levels [66,106]. Importantly, combinatory protocols for PCa treatment, including autophagy inhibitors, led to both enhancement of the pro-apoptotic actions of OGX-427 and hampering of tumor growth. These findings strongly suggest that autophagy may be a compensatory response for HSP27 depletion [243].

In fact, OGX-427 has been used as monotherapy or in combination with docetaxel or prednisone in clinical trials including patients with mCRPC [243] (Table 3). When compared to controls, patients treated with either OGX-427 alone or OGX-427/docetaxel showed diminished PSA levels [244,245]. On the other hand, combination therapy including OGX-427 and prednisone showed better results in terms of PSA level decline and delay of progression [245]. Another important aspect of OGX-427 usage in PCa treatment is that it has a good safety profile with minimal side effects that usually accompany such drugs like dizziness, hemolytic uremic syndrome, and pulmonary embolus [19]. Despite its promising activities in vitro and in clinical trials, OGX-427 usage as monotherapy in clinical practice is not widely implemented. Due to the drawback of other systemic actions that reduce its effectiveness, OGX-427 is more commonly being used in combination therapies with HSP90 inhibitors or PTEN targeting agents [93,130].

Recently, HSP27 targeting approaches involving its oligomerization have emerged. Small molecules such as zerumbone (ZER) caused altered dimerization of HSP27 by cross-linking its cysteine residues leading to inhibition of oligomerization and consequent sensitization to radiation therapy [246,247]. In non-small cell lung cancer (NSCLC), J2 compound that is highly cross-linked to HSP27, sensitized HSP27 highly expressing lung cancer cells to HSP90 inhibitors, taxol, and cisplatin [248]. Using computational drug repositioning approach, Heinrich et al., could identify six compounds, five of which could bind HSP27 and limit its chaperoning activity. Strikingly, in vitro studies have revealed that these compounds attenuate the development of drug resistance in cancer cell lines [249]. Such approaches may be applicable in future studies to monitor PCa.

### 4.5. Clusterin

Although not obviously classified as HSP, clusterin is a molecular chaperone that shares several characteristics with classical small HSPs [250]. This secretory heterodimeric glycoprotein is highly conserved among species and expressed in almost all tissues [251]. Like many HSPs, the expression of clusterin is increased in response to cellular stress where it serves to bind target proteins preventing their aggregation [8,252]. Clusterin is initially synthesized as 70–80 kDa polypeptide precursor that is cleaved into two 40 kDa α and β subunits, which later assemble to form a functional heterodimer connected by disulphide bridges (Figure 2, lower panel) [10,253,254]. The name “clusterin” was firstly used to reflect the ability of this molecule to cluster numerous cell types [251]. Later, it was identified to contribute to many crucial physiological and pathological processes including apoptosis, DNA repair, cell cycle control, cell adhesion, lipid transport, immunological functions, and carcinogenesis [8,251,253,254].

Expression of clusterin is, like other HSPs, regulated by HSR in which HSF1/HSF2 heterocomplex bind to CLU-specific promoter element to initiate its transcription [255]. Moreover, besides other transcription factors, androgens can also control its transcription through binding to ARE in the promoter region of the encoding gene [256]. Importantly, Y-box binding protein-1 (YB-1) is another transcription factor that regulates clusterin expression and is strongly involved in therapeutic resistance of PCa to paclitaxel and enzalutamide treatments [97,257].

#### 4.5.1. Biological Functions of Clusterin in Prostate Cancer

Overexpression of clusterin has been reported in several malignant tumors such as breast, lung, liver, kidney, colorectal, and prostatic tumors [230]. In rat prostatic tissue, clusterin was primarily identified as testosterone-repressed prostate message 2’ (TRPM-2), linking its high expression to regression of rat prostate. However, later research showed that clusterin is more likely stress or apoptosis correlated rather than being an androgen suppressed gene [19,131]. In PCa cells, clusterin has been looked at as a key molecule supporting resistance to chemotherapy, ionizing radiation, and protection against oxidative stress or other cellular stresses [19].

The multiple functions of clusterin in PCa are summarized in (Table 2). Clusterin attaches to the apoptosis regulator BAX (Bcl2-L-4) and prevents its oligomerization thus hampering caspase signaling and chemotherapy induced apoptosis [94]. Moreover, it has been reported that simultaneous activation of clusterin and Akt is associated with resistance to PCa apoptotic drugs such as docetaxil. Furthermore, both molecules were found to be reciprocally regulated, whereby one molecule activates the other [95,96]. Within the MAPK pathway, clusterin and Akt have been shown to withstand the action of androgen receptor antagonist enzalutamide through phosphorylation of YB-1 [97]. It is worth noting that, together with GRP78 [98], clusterin was found to assist proteostasis by participating in the ER stress and unfolded protein response mechanisms [230]. In this context, binding of clusterin to misfolded proteins and its retro-translocation from the ER to the cytoplasm relief the ER protein aggregates and counteract apoptosis [230]. Similar to secretory HSP90, clusterin is a key factor of the epithelial-to-mesenchymal transition (EMT) in PCa. It functions downstream of TGFβ and Twist1 [107,108] where high expression of clusterin confers Smad2–Smad3 stability and potentiates TGFβ-mediated Smad transcriptional activity [258]. On the other hand, clusterin depletion inhibited EMT through slug downregulation [259]. In addition, clusterin aids in the development of pro-angiogenic criteria in PCa cells [111].

#### 4.5.2. Prognostic Value of Clusterin in Prostate Cancer

Several observations link clusterin expression to the severity or pathological grade of PCa. Clusterin expression is increased in mice lacking the tumor suppressor *Nkx3-1* gene in the early stages of the prostate tumor [260]. Also, high levels of clusterin were detected in both biopsy and radical prostatectomy (RP) samples from patients showing pre-operative high PSA or high pathological grade values [10,261]. In malignant prostatic neoplasms, androgen ablation led to significantly elevated levels of clusterin underlining its role in therapeutic resistance and generalized protection of PCa cells [262]. Moreover, in patients subjected to radical prostatectomy (RP), clusterin has been used as a biochemical marker for disease recurrence [10,263]. Additionally, clusterin levels were significantly elevated following neoadjuvant hormone therapy [10] and in patients having high-grade PCa with extracapsular extension [264]. Taken together, data obtained in this respect consider clusterin as a biomarker for high grade of the disease, post-treatment stress, and poor prognosis [19].

#### 4.5.3. Targeting Clusterin in Prostate Cancer

Similar to HSP27, targeting clusterin with small molecule inhibitors is not accessible because the ATP binding site is not there. Therefore, antisense oligonucleotides and siRNA technologies have been developed to inhibit the carcinogenic propensities of clusterin [131]. In this regard, suppression of clusterin in prostate tumor models with antisense oligonucleotides resulted in increased cell death, delayed tumor growth, and reduced metastasis [108,131,132]. Combined therapeutic approaches—including targeting clusterin with antisense oligonucleotides or siRNA in addition to other drugs like docetaxel, paclitaxel, mitoxantrone—lead to enhanced drug efficacy and increased sensitivity of PCa cells to ionizing radiation [132,135]. Furthermore, clusterin antisense oligonucleotides have maximized the efficacy of the AR antagonist enzalutamide [97], leading to rapid degradation of the AR in YB-1 and FKBP52 mediated pathway [19]. In mCRPC patients undergoing radical prostatectomy, the clusterin antisense oligonucleotide OGX-011 was administered in a Phase I study on a weekly basis. When combined with neo-adjuvant androgen deprivation therapy [265], this regimen showed reduction of clusterin expression in PCa cells and lymph node tissues. 

#### 4.5.4. Targeting Clusterin within Combination Therapies for mCRPC

The fact that HSPs like HSP90 and 70 are co-overexpressed with clusterin in mCRPC makes them ideal molecules to be co-targeted with clusterin in mCRPC treatment. This is because non-targeted overexpressed chaperone candidates have been believed to reduce the efficacy of certain drugs dealing with or targeting only one molecule. The problem of therapeutic resistance to HSP90 inhibitors have been reported in cases where clusterin has increased its expression [139]. As a support for this notion, increased expression of clusterin has been shown to promote HSF1 transcriptional activity, while clusterin silencing—together with HSP90 inhibitor—could significantly inhibit HSF1 mediated transcription [137] 

In other treatment protocols, clusterin has been involved in combination therapies in many clinical trials of mCRPC. In a Phase II clinical trial, dual co-administration of docetaxel and prednisone besides OGX-011 resulted in better survival rates in mCRPC patients compared to patients receiving docetaxel and prednisone alone [266]. However, there were some controllable toxic side effects related to OGX-011 which included fever, rigor, diarrhea, and rash [19,266]. 

Nevertheless, the promise of utilizing OGX-011 in combination therapies for metastatic PCa has extended to include Phase III trials like those of the AFFINITY and SYNERGY trials, where OGX-011 was added to either cabazitaxel or docetaxel, respectively [19,267,268]. Though recent results of the two studies showed no survival benefit from OGX011 inclusion in the treatment protocols, there is still hope for finding the perfect combination therapy in mCRPC [267,268]. Collectively, data coming from those clinical trials, as well as other similar studies testing combination therapies, may help in determining the optimal dose and regimen of OGX-011 in patients with mCRPC [19].

## 5. Conclusions, Challenges, and Perspectives

HSPs represent attractive targets for the treatment of PCa, especially in mCRPC. Not only can HSPs fine-tune the AR transcription and protein levels, but also they chaperone many oncoproteins making their targeting beneficial in many aspects. Though increasing auspicious trials in the course of HSPs inhibition as a tool for PCa treatment, identifying the optimum protocol involving HSP inhibition has not been successful to date. This is due to little efficacy or undesirable toxic side effects in most clinical trials. Therefore, novel therapeutic strategies and highly selective HSP inhibitors with potent effects are in demand to achieve the optimum therapeutic outcomes for PCa patients.

## Figures and Tables

**Figure 1 cancers-11-01194-f001:**
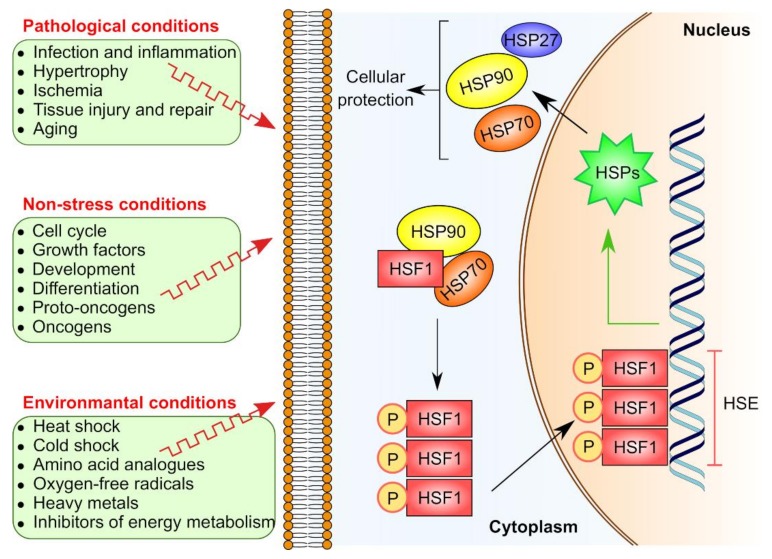
Induction of HSR by HSF1. Various factors including environmental, non-stress and pathological conditions can trigger HSP expression via HSF1 pathway. In non-activated state, HSF1 is sequestered in the cytoplasm due to its binding to chaperone complex including HSP70 and HSP90, thus prevented from performing its transcriptional activity. Upon activation, the chaperone complex dissociates and HSF1 is liberated, homotrimerizes and translocated to the nucleus. In the nucleus, the HSF1 homotrimer binds to the heat shock elements (HSEs) that are located upstream to heat shock gene promotors to initiate the transcription of its target genes including HSP genes.

**Figure 2 cancers-11-01194-f002:**
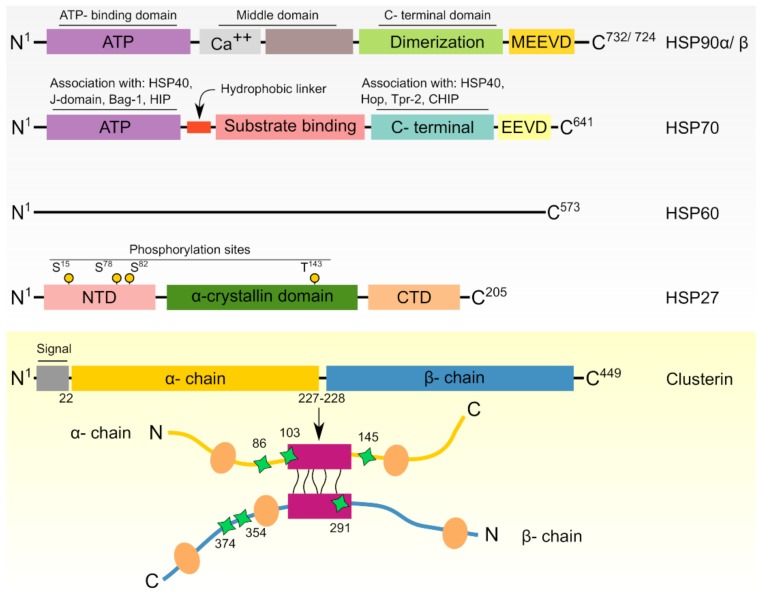
Domain architecture of common chaperones involved in PCa. From top to bottom, human HSPs including HSP90α/β, HSP70, HSP60, HSP27, are displayed relative to their relative length where the number of amino acids constituting each chaperone is written as superscripts in their C-terminals. HSPs have N- and C-terminal domains in addition to middle domain like HSP90 and HSP70 while linear representation of human HSP60 domains is still unclear due to its oligomerization and complex association with HSP10. HSP27 has a middle highly conserved α-crystalline domain. Phosphorylation sites of HSP27 are designated as yellow spheres representing either phosphorylated serine or threonine amino acid residues. Human clusterin (CLU) exists initially as polypeptide precursor which undergoes proteolytic cleavage of its first 22 amino acid secretory signal in addition to its cleavage at Arg227–Ser228 to yield two chains; namely α and β chains. The two chains are arranged in anti-parallel orientation to constitute a heterodimeric molecule. Pink boxes represent cysteine-rich centers that are linked by five disulfide bridges. Yellow ovals point to predicted amphipathic α-helices while green tetragonal stars refer to the N-glycosylation sites.

**Figure 3 cancers-11-01194-f003:**
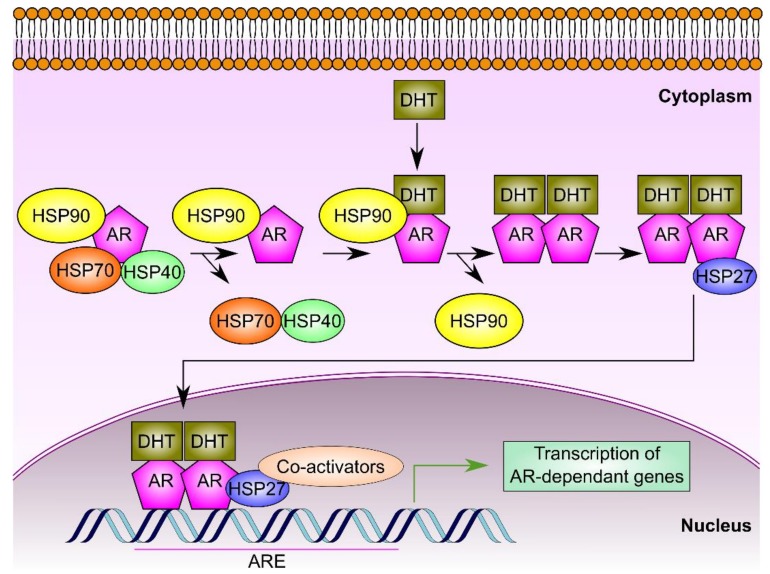
HSPs regulate AR signaling. A complex of chaperones including HSP90 associates with the AR to expedite and maintain its high affinity binding conformation, thus allowing for DHT interaction. As a consequence, the AR forms a dimer and the chaperone complex dissociates. Thereafter, HSP27 binds to the AR homodimer enabling its nuclear translocation, subsequent binding to the ARE, and activation of transcription of AR-dependent genes.

**Figure 4 cancers-11-01194-f004:**
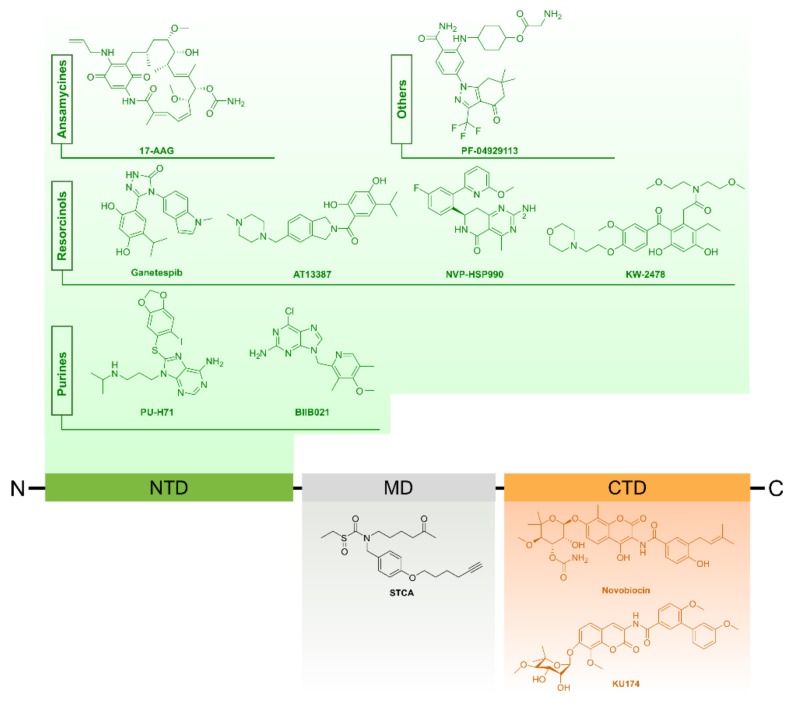
Common small inhibitors of HSP90.

**Table 1 cancers-11-01194-t001:** Variant families of HSPs and sample members from each [26].

Family Name	Sample Protein Members	Gene Name/M.W (kDa)	Cellular Location	Co-Chaperones	Roles	Citation
Small HSPs	HSP10	*HSPE1*/10	Mitochondria	None	Molecular chaperone (co-factor for HSP60)	[27]
	HSP27	*HSPB1*/22	Cytosol/nucleus		Molecular chaperone	[28,29,30]
HSP40/DNAJ	HSP40	*DNAJB1*/38	Cytosol	None	Molecular chaperone (co-factor for HSP70)	[31,32,33]
	Tid1	*DNAJA3*/Isoform 1/52	Cytosol			
		*DNAJA3*/Isoform 2/49	Mitochondria			
HSP60	HSP60	*HSPD1*/61	Cytosol, mitochondria, chloroplast	HSP10	Chaperonin	[34,35]
HSP70	HSP70	*HSPA1A*/70	Cytosol	HSP40, Grpe, Bag1, Bag3, Hip, Hop, CHIP	Molecular chaperone	[36,37]
	HSP70-2	*HSPA1B*/70	Cell surface			
	HSC70	*HSPA8*/71	Cytosol			
	GRP75/Mortalin	*HSPA9*/73	Mitochondria			
	GRP78	*HSPA5*/72	ER			
HSP90	HSP90A	*HSPC1*/86	Cytosol	P23, Aha1, Cyp40, Cdc37, Hop, FKBP51, FKBP52	Molecular chaperone	[38,39,40]
	HSP90B	*HSPC3*/84	Cytosol			
	GRP94	*HSPC4*/92	ER, cytosol			
	TRAP1	*HSPC5*/75	Mitochondria			
Large HSPs	HSP110	*HSP110*/96	Cytosol	None	Holdase, molecular chaperone	[41,42,43]
	GRP170	*HYOU1*/170	ER			

**Table 3 cancers-11-01194-t003:** Common HSP targeting therapies in prostate cancer.

HSP Inhibitor	Molecular Mechanism	Used Cell Line/Model	Reference
Targeting HSP90			
17-AAG	Interferes with nuclear translocation of AR, postpones castration resistance, and promotes cell viability	LuCaP35 xenograft model	[124]
	Downregulates the levels of HSP90 clients including AR, ERBB2, ERBB3, and Akt, and inhibits PCa growth	CWR22R & CWRSA6 xenografts	[84]
Ganetespib	Reduces the expression of HSP90 client proteins such as AR, Akt, ERK, IGFR-1, EGFR, and STAT3, enhances apoptosis and interferes with growth of PCa cells	PC3 and 22Rv1 xenografts	[81]
NVP-AUY922	Decreases HSP90 client proteins like ERBB2, c-RAF, CDK-4, Akt, and HIF-1α, inhibits growth of PC3 xenografts, and suppresses lymphatic metastasis	PC3LN3 orthotopic lymph node mPCa model	[102]
	Exhibits anti-proliferative and pro-apoptotic functions	ex vivo model of primary PCa	[82]
AT13387	Causes degradation of HSP90 client proteins such as AR, Akt, ERBB2, and c-RAF, hinders AR nuclear translocation and inhibits its transcriptional activity and displays anti-proliferative and growth limiting effects in vivo	VCaP, LNCaP, 22Rv1	[79]
NVP-HSP990	Performs anti-proliferative and pro-apoptotic functions	ex vivo model of primary PCa	[82]
PF-04929113 (SNX-5422)	Depletes HSP90 client proteins like AR, ERBB2, Akt, and ERK, suppresses RANKL-mediated osteoclast differentiation interferes with growth of PCa xenografts	LNCaP xenografts	[86]
Gamitrinibs	Exhibits pro-apoptotic effects, reduces growth of PCa xenografts and inhibits bone metastasis	PC3 xenografts and orthotopic model of PCa	[128]
Shepherdin	Decreases HSP90 client proteins levels including survivin, Akt, CDK-4, and CDK-6, displays pro-apoptotic effects and delays growth of PCa xenografts	PC3 xenografts	[83]
KU174	Supports apoptosis, delays growth of PCa xenografts, and depletes HSP90 client proteins, such as AR, survivin, ERBB2, and Akt	PC3 xenografts	[80]
Targeting HSP70			
MKT-077	Delays growth of PCa cells	DU145 xenografts	[129]
Targeting HSP27			
OGX-427	Interferes with HSP90-AR binding, promotes AR proteasomal degradation, supports apoptosis, suppresses growth of PCa xenografts, reduces serum PSA levels	LNCaP xenografts	[66]
	Hinders metastasis	PC3M model of mPCa	[106]
Targeting HSP90 and HSP27			
Combination of PF-04929113 and OGX-427	The anti-proliferative and pro-apoptotic actions of PF-04929113 are augmented by OGX-427 due to its synergetic effect resulting in PCa growth inhibition	LNCaP xenograft	[130]
Targeting clusterin			
Clusterin antisense oligonucleotide	Enhances apoptosis and counteracts recurrence in castration-sensitive PCa	Shionogi rat prostate tumors	[131]
	Potentiates paclitaxel efficacy	Shionogi rat prostate tumors	[132,133]
	Increases efficacy of paclitaxel or mitoxantrone	PC3 xenografts	[134]
	Enhances response to radiation therapy	PC3 xenografts	[135]
OGX-011	Anti-metastatic effect	PC3M model	[108]
	Enhances efficacy of enzalutamide	LNCaP xenografts	[97]
	Increases sensitivity to paclitaxel or mitoxantrone	PC3 xenografts	[136]
Targeting HSP90 and clusterin			
Combination of 17-AAG, PF-04929113 and OGX-011	OGX-011 enhances apoptotic and anti-proliferative actions of 17-AAG and PF-04929113 resulting in synergistic inhibition of PCa growth	PC3 and LNCaP xenograft	[137]

**Table 4 cancers-11-01194-t004:** Key HSP90 co-chaperones and their involvement in prostate cancer.

Co-Chaperone	Roles	Targeting Approach	References
Cdc37	Controls the HSP90 ATPase cycle by assisting the recruitment of kinase client proteins to the Hsp90 machinery	siRNA and natural product inhibitors (celastrol, withaferin A and taxifolin)	[121,161,162,163,164,165]
	Interacts with Vav3, a co-activator of AR transcriptional activity, thus, increasing prostatic cell proliferation		
p32	Activates AR transcription by facilitating the binding of AR to the androgen response element (ARE) Participates in steroid receptor folding and signaling	Gedunin (inhibitor)	[119,166]
	Mediates the binding of AR to HSP90 by acting as a bridge between the two molecules		
Immunophilin FKBP52	Stabilizes Hsp90-AR interaction	Knockout and inhibition by MJC13	[64,126,167]
SGTA	Interacts with AR and keeps it in the cytoplasm	Knockdown and knockout	[168,169,170]
	Enhances PCa proliferation and survival by promoting Akt signaling pathway		

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
