# Peer review of "The Multiple Roles and Therapeutic Potential of Molecular Chaperones in Prostate Cancer"

_cancers, 2019, doi:10.3390/cancers11081194_

Round 1

Reviewer 1 Report

This review article provides some interesting information as well as some novel insight into the complexity of the implication of chaperones in PCa. The authors discuss in depth the characteristics of the members of this protein superfamily, their correlation with PCa development and progression and, eventually, the pharmacological approaches currently available for PCa. I found it a very comprehensive review of the literature with extensive relevant references and includes a lot of figures and useful tables for illustration. I have no hesitation in recommending that it be accepted for publication after a few typos and other minor details have been attended to.

I have only a few misspellings and minor comments that should be corrected or considered before publication:

- The authors provide a useful “abbreviation list” at the end of the manuscript. However, they should use the abbreviation “PCa” throughout the manuscript. Thus, please in Lines 71, 74, 81, 128…etc “prostate cancer” should be replaced with “PCa”

- Line 91: “All” should be “all”

- Line 183: 90kDa should be 90 kDa. The same in line 530 “30µM” and in line 643 “40kDa”

- Line 338: In other studies,, (doble commas)

- Line 353: “inlcudes"

Author Response

We thank both reviewers for their positive assessment of our manuscript, their suggestions and constructive comments. In the revised version of the manuscript, we have addressed all the points raised and made the changes accordingly.

Reviewer # 1

This review article provides some interesting information as well as some novel insight into the complexity of the implication of chaperones in PCa. The authors discuss in depth the characteristics of the members of this protein superfamily, their correlation with PCa development and progression and, eventually, the pharmacological approaches currently available for PCa. I found it a very comprehensive review of the literature with extensive relevant references and includes a lot of figures and useful tables for illustration. I have no hesitation in recommending that it be accepted for publication after a few typos and other minor details have been attended to.

I have only a few misspellings and minor comments that should be corrected or considered before publication:

- The authors provide a useful “abbreviation list” at the end of the manuscript. However, they should use the abbreviation “PCa” throughout the manuscript. Thus, please in Lines 71, 74, 81, 128…etc “prostate cancer” should be replaced with “PCa”

- Line 91: “All” should be “all”

- Line 183: 90kDa should be 90 kDa. The same in line 530 “30µM” and in line 643 “40kDa”

- Line 338: In other studies,, (doble commas)

- Line 353: “inlcudes"

Response: All comments have been addressed in the revised version and the corrections were made according to the reviewer's suggestions.

Reviewer 2 Report

The manuscript is well written and provides valuable information to add to our current understanding of heat shock proteins and other chaperons, mainly Clusterin, in prostate cancer (PCa) pathogenesis and androgen resistance. The discussed literature supports the authors conclusions in a logical manner and the figures and tables are presented well. However, the following are some suggestions that may further improve the manuscript quality:

The authors have briefly mentioned the challenges of targeting HSP in PCa- this being a topic of utmost importance could be discussed in more detail including the side effects, toxicity, and the need for more specific and selective targeted therapies with regard to HSP inhibitors. Another topic of relevance in PCa that would add to the content of the review article is the role of HSPs in neuroendocrine PCa.

Here are some suggested articles that could be discussed:

References -1. Neuroendocrine prostate cancer (NEPCa) increased the neighboring PCa chemoresistance via altering the PTHrP/p38/Hsp27/androgen receptor (AR)/p21 signals (Oncogene. 2016).

References -2. The neuroendocrine-derived peptide parathyroid hormone-related protein promotes PCa cell growth by stabilizing the androgen receptor (Cancer Res. 2009).

The following are some minor suggestions:

Section 2, the first sentence needs to be grammatically corrected- “Though their important and numerous functions in cellular homeostasis” Abbreviation of some words (e.g. HSF, HSFY) should be used at first, and followed-up subsequently. In Table 2, HSPs function e.g. Anti-apoptosis, AR-trafficking stability, invasion and metastasis need to be merged in respective rows.  And the same should be followed in the table. At some places PC3 and PC-3 are written and needs to be corrected, also at several places extra spaces between two words should be removed. Table 4 should be edited- Molecules and related functions should be kept in same row. Texts on different pages (11-13,19, 22 etc.) are out of bottom margin and need to be adjusted. Subheads for 4.4.1 a/b; 4.1.2 a/b/c/d;  4.4.2 A/B/c/d should be followed as per the journal instructions. 

Author Response

We thank both reviewers for their positive assessment of our manuscript, their suggestions and constructive comments. In the revised version of the manuscript, we have addressed all the points raised and made the changes accordingly.

Reviewer # 2

The manuscript is well written and provides valuable information to add to our current understanding of heat shock proteins and other chaperons, mainly Clusterin, in prostate cancer (PCa) pathogenesis and androgen resistance. The discussed literature supports the authors conclusions in a logical manner and the figures and tables are presented well. However, the following are some suggestions that may further improve the manuscript quality. The authors have briefly mentioned the challenges of targeting HSP in PCa- this being a topic of utmost importance could be discussed in more detail including the side effects, toxicity, and the need for more specific and selective targeted therapies with regard to HSP inhibitors.

Response: We agree with the reviewer that there are several challenges for targeting HSPs in PCa. At the same time, we think that detailed discussion of this part including side effects, toxicity ..etc, will take the manuscript into a more clinical aspect, which is not the primary focus of this review. We sought primarily in the current review to discuss the molecular roles and cellular mechanisms of HSPs in PCa and their potential use in PCa therapy. In this context, we believe that the information regarding side effects and toxicity that are included in this manuscript are adequate. Other clinical trials or studies including clinical outcomes and side effects due to targeting HSPs in PCa can be discussed in another related topic which would be more clinically oriented.

Another topic of relevance in PCa that would add to the content of the review article is the role of HSPs in neuroendocrine PCa.

                Here are some suggested articles that could be discussed:

                References

Neuroendocrine prostate cancer (NEPCa) increased the neighboring PCa chemoresistance via altering the PTHrP/p38/Hsp27/androgen receptor (AR)/p21 signals              (Oncogene. 2016). The neuroendocrine-derived peptide parathyroid hormone-related protein promotes PCa cell growth by stabilizing the androgen receptor (Cancer Res. 2009).

Response: In the revised version of the manuscript, we included a new paragraph regarding the role of HSPs in HSPs in neuroendocrine PCa (Lines 206-214 and 594-601).

The following are some minor suggestions:

                - Section 2, the first sentence needs to be grammatically corrected- “Though their     important and numerous functions in cellular homeostasis”

Response:All other comments have been addressed in the revised version and the corrections made according to the reviewer's suggestions.
